# Superconductivity in kagome metals due to soft loop-current fluctuations

**Daniel J. Schultz** [1] ✉, **Grgur Palle** [2,3], **Asimpunya Mitra** [4], **Yong Baek Kim**[4], **Rafael M. Fernandes** [2,3] & **Jörg Schmalian** [1,5]

We demonstrate that soft fluctuations of translation symmetry-breaking loop currents provide a mechanism for unconventional superconductivity in kagome metals that naturally addresses the multiple superconducting phases observed under pressure. Focusing on the rich multi-orbital character of these systems, we show that loop currents involving both vanadium and antimony orbitals generate low-energy collective modes that couple efficiently to electrons near the Fermi surface and mediate attractive interactions in two distinct unconventional pairing channels. While loop-current fluctuations confined to vanadium orbitals favor chiral $d + id$ superconductivity, which spontaneously breaks time-reversal symmetry, the inclusion of antimony orbitals stabilizes an $s^{\pm}$ state that is robust against disorder. We argue that these two states are realized experimentally as pressure increases and the antimony-dominated Fermi surface sheet undergoes a Lifshitz transition.

Kagome metals of the family $AV_3Sb_5$, with A = Cs, Rb, and K, exhibit a rich phase diagram featuring charge density-wave (CDW) order and superconductivity (SC) as well-established symmetry-broken states[1–24]. Moreover, they have emerged as prime candidates for hosting loop-current (LC) states that break both time-reversal symmetry and translational invariance, suggesting that one phase may facilitate the emergence of the other (for a recent review, see ref. [25]). In this work, we demonstrate that fluctuations of loop currents or, equivalently, the fluctuations of orbital-magnetic fluxes, give rise to unconventional superconductivity in the kagome metals.

The nature of the CDW-induced lattice distortions has largely been clarified through scanning tunneling microscopy[6,26–28] and X-ray scattering experiments[1,29–37]. Theoretically, they have been proposed to arise from the interplay between phonons and electronic states which are primarily made of vanadium $3d$ orbitals, with additional contributions coming from the antimony $5p$ states[38–40]. The LC state, on the other hand, has often been tied to the Van Hove singularities near the Fermi energy whose states are dominated by vanadium $3d$ orbitals[41–53]. Interestingly, SC seems to be closely related to a Fermi surface sheet dominated by antimony $5p$ states, as its suppression via a

Lifshitz transition coincides with a sudden drop in the transition temperature[54,55]. Given the distinct role of these orbital states, the link between CDW and LC states, on the one hand, and pairing, on the other, is an open problem. One possible answer is that CDWs and SCs are both driven by strong electron-phonon interactions[56–59]. Observations that support some form of time-reversal symmetry-breaking in the SC state[60,61] and the fact that electron-phonon interactions do not tend to support sufficiently strong LC states, however, suggest that purely electronic interactions are important.

There are robust experimental reports of time-reversal symmetry-breaking inside the CDW state[17,27,60,62–68], although a net magnetization is unlikely to be present[69]. Nevertheless, it is unclear whether LCs form a stable long-range order at zero magnetic field, or whether they remain fluctuating[70] and condense only in the presence of a magnetic field, with an onset that either coincides with the CDW transition or occurs at a lower temperature[66]. Either way, this broad set of experimental observations strongly supports a scenario in which LCs are present as soft collective excitations. Their fluctuations should then play a role in the low-energy physics of the kagome metals. The goal of this paper is to analyze the consequences of these fluctuations. As we

[1]Institute for Theoretical Condensed Matter Physics, Karlsruhe Institute of Technology, Karlsruhe, Germany. [2]Department of Physics, The Grainger College of Engineering, University of Illinois Urbana-Champaign, Urbana, IL, USA. [3]Anthony J. Leggett Institute for Condensed Matter Theory, The Grainger College of Engineering, University of Illinois Urbana-Champaign, Urbana, IL, USA. [4]Department of Physics, University of Toronto, Toronto, ON, Canada. [5]Institute for Quantum Materials and Technologies, Karlsruhe Institute of Technology, Karlsruhe, Germany. ✉e-mail: daniel.schultz@kit.edu

shall see, this requires detailed knowledge of the complex electronic structure.

The electronic structure of the AV$_3$Sb$_5$ systems consists of multiple bands made up of vanadium (V) and antimony (Sb) states[1,71-75]; see Figs. 1, 2. The V $3d$ states play an important role near the three saddle points $M_\ell$, $\ell \in \{1, 2, 3\}$. These points are connected by three wave vectors $Q_\ell$ that coincide with the in-plane components of the CDW modulation (e.g., $Q_3 = M_2 - M_1$; see Fig. 3f), highlighting the crucial role played by these electronic degrees of freedom in the formation of the CDW state, in combination with lattice degrees of freedom. Sb states, frequently ignored in low-energy models of the kagome metals, are important in making the $M$-point saddle points cross the Fermi level at non-zero $k_z$[56] and also form a separate Fermi surface sheet near the $\Gamma$-point. The states from the apical and planar Sb atoms were recently shown to be significant in microscopic models of LC and CDW order[40,52,76]. Moreover, they are also closely tied to superconductivity since the pressure-induced suppression of this antimony Fermi surface sheet via a Lifshitz transition[54,77,78] coincides with the disappearance of superconductivity[79,80]. Interestingly, another superconducting state is then observed at higher pressure[79], even without this Sb-Fermi pocket. In addition, quasi-particle interference spectroscopy of KV$_3$Sb$_5$ finds evidence that the superconducting gap on the Sb-Fermi surface sheet is comparatively large[67]. This suggests that, at least at low pressure, Sb states are important in determining the superconducting properties. Given the complexity of the electronic structure and the nature of

competing states, a theory for the mechanism of superconductivity must go beyond simple model descriptions and properly account for the symmetry and the microscopic nature of symmetry-broken and fluctuating states.

In this paper, we demonstrate that LC fluctuations give rise to two dominant unconventional superconducting states depending on the system's paramaters: $s^\pm$ pairing with a large SC gap at the Sb-Fermi surface, and a chiral $d + id$ state that breaks time-reversal symmetry. These results follow from a microscopic model that fully incorporates the multi-orbital character of the kagome lattice, including the 30 V-$3d$ and Sb-$5p$ orbitals per unit cell[81]. We then carry out a comprehensive symmetry classification of LC states, identifying allowed LC patterns consistent with the lattice and time-reversal symmetry-breaking. Given the crucial role of Sb states in shaping low-energy fluctuations, we extend the conventional treatment, which considers LCs confined to V sites, to include LC patterns that traverse between V and Sb orbitals. The key result is that the nature of the SC state is determined by which of these two types of symmetry-equivalent LC patterns displays the strongest fluctuations. LCs involving only V orbitals favor $d + id$ pairing, whereas LCs involving both V and Sb orbitals favor $s^\pm$ pairing. We identify these two distinct pairing states with the two separate SC states observed experimentally in CsV$_3$Sb$_5$ as pressure is increased[79]. To avoid complications related to the reconstruction of the Fermi surface in the CDW/LC ordered state, we focus on the regime without long-range order. This corresponds to the $r > 0$ region of Fig. 1c, which

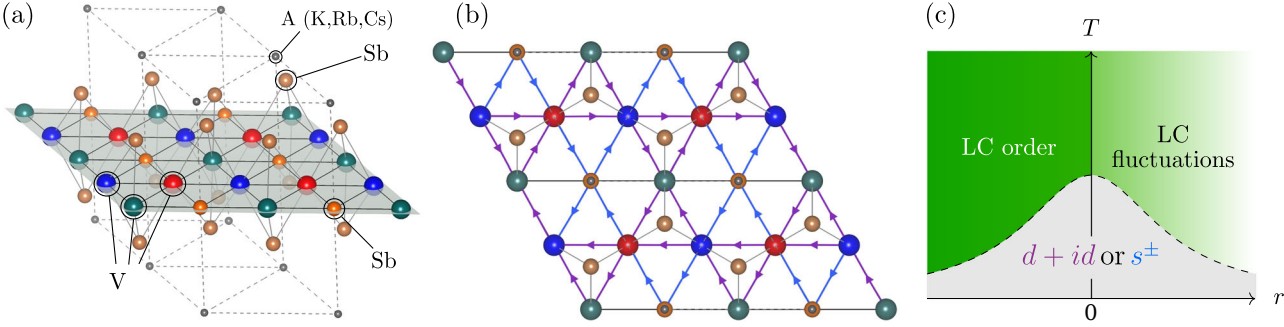

**Fig. 1 | Crystal structure and summary of the main result. a** The crystal structure of the AV$_3$Sb$_5$ kagome metals. **b** Top-down view of the crystal structure, showing the V-kagome plane (red, blue, teal), the triangular lattice of planar Sb atoms (orange; below small gray circles), the honeycomb lattice of (brown) apical Sb (which live both above and below the V-kagome plane), and the triangular lattice of the alkali metal A (K,Rb,Cs) which live directly above the in-plane Sb. The atom sizes are not to scale and (K,Rb,Cs) have been shrunk to appear smaller than Sb for

visualization purposes. **c** A schematic phase diagram of our envisioned scenario. $T$ is temperature and $r$ is a tuning parameter (e.g., pressure). For $r > 0$ the system is disordered, yet loop-current fluctuations can still be soft and mediate Cooper pairing. It is this superconductivity that we are studying in the current work. V-V loop currents (purple arrows in (**b**)) drive $d + id$ pairing, whereas V-Sb loop currents (blue arrows in (**b**)) drive $s^\pm$ pairing.

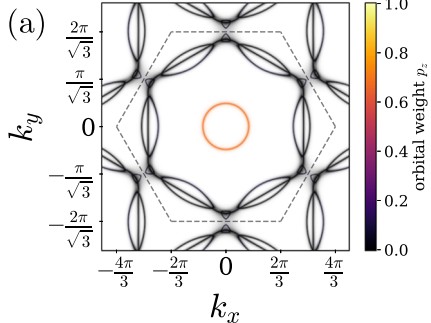
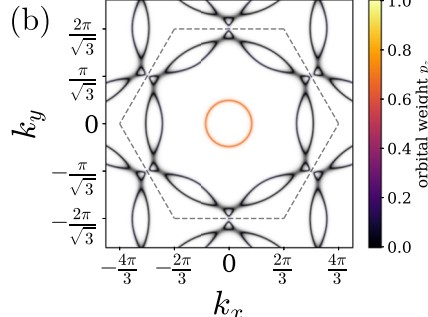

**Fig. 2 | Fermi surface of realistic tight-binding models.** The Fermi surfaces of **a** the 30 band model and **b** the 13 band model, consisting only of orbitals (or linear combinations thereof) which are odd under $\sigma_h: z \mapsto -z$. The key difference between **a** and **b** is that **a** has an additional large hexagonal Fermi surface whose vertices almost reach the **M**-points in the Brillouin zone, which is due to orbitals even under

$\sigma_h$. The remaining features of importance are the Van Hove singularities at the **M**-points, which the Fermi surface comes close to. The electrons near the **M**-points are primarily made of $d_{zx}$, $d_{yz}$ vanadium orbitals. There is also a circular pocket around the $\Gamma$-point, arising primarily from $p_z$ orbitals of planar Sb atoms (orange shading).

can be experimentally achieved by suppressing the long-range CDW/LC order with pressure[34,54,82–86] or doping[55,87,88].

LC fluctuations have also been proposed as a pairing glue in the cuprates[89–93]. Recent work[94,95], which shows that they are not effective at driving superconductivity, appears at first glance to contradict our results. However, as we explain later, the crucial distinction that allows soft LC fluctuations to efficiently induce Cooper pairing in the kagome systems is that they break translation symmetry, in which case the reservations of refs. 94,95 no longer apply.

## Results

### Electronic structure

To describe the electronic structure of AV$_3$Sb$_5$, we use the 30-band tight-binding model of ref. 81. This model includes five V-3$d$ states at the three kagome sublattices in the unit cell and three Sb-5$p$ states at five locations. In our analysis, we focus on the two-dimensional states with $k_z = 0$, thereby accounting for the anisotropic electronic structure of AV$_3$Sb$_5$. We may then split the orbitals into those that are even and those that are odd under horizontal reflections $\sigma_h: z \mapsto -z$ with respect to the V-kagome plane (Fig. 1a). The motivation for this separation is that microscopic theories of LC order find that they are naturally formed of those orbitals that are odd under $\sigma_h$[52,81]. In Fig. 2, we show the resulting Fermi surface of the full tight-binding model (a) and of the mirror-odd subsector (b).

### Classification of loop-current states

The LC patterns we classify according to two distinct properties: their symmetry transformations under the crystallographic space group and their orbital compositions (i.e., between which types of orbitals do the LCs flow). Broken symmetry states can be classified according to the irreducible representations (irreps) of the space group of the parent disordered state. While various Ginzburg-Landau theories[39,43,47,96], as well as more microscopic symmetry analyses[46,49], have been carried out for charge and LC order in kagome systems, we proceed with an approach that, in addition, takes into account the 3$d$ V and planar 5$p$ Sb orbital structure. This allows us to explicitly construct the couplings between the LC patterns and the electronic structure described in the previous section. We consider all possible LC patterns that can exist within a 2 × 2 unit cell and that break translation invariance with ordering vectors $\mathbf{Q}_{\ell=1, 2, 3}$ which connect the Van Hove points (Fig. 3). Regarding the orbital compositions, there are two different categories we consider: LC patterns that impact the phase of nearest-neighbor hopping parameters between V sites, and LC patterns that impact the phase of the hopping parameters between V sites and planar Sb sites. In both of these scenarios, only the 13 orbitals that are odd under $\sigma_h$ are involved. We will later show that the orbital composition is what selects between $d + id$ vs. $s^\pm$ pairing.

Using standard group theory techniques[47,49,97], we classify these patterns according to the irreducible representations of the reduced

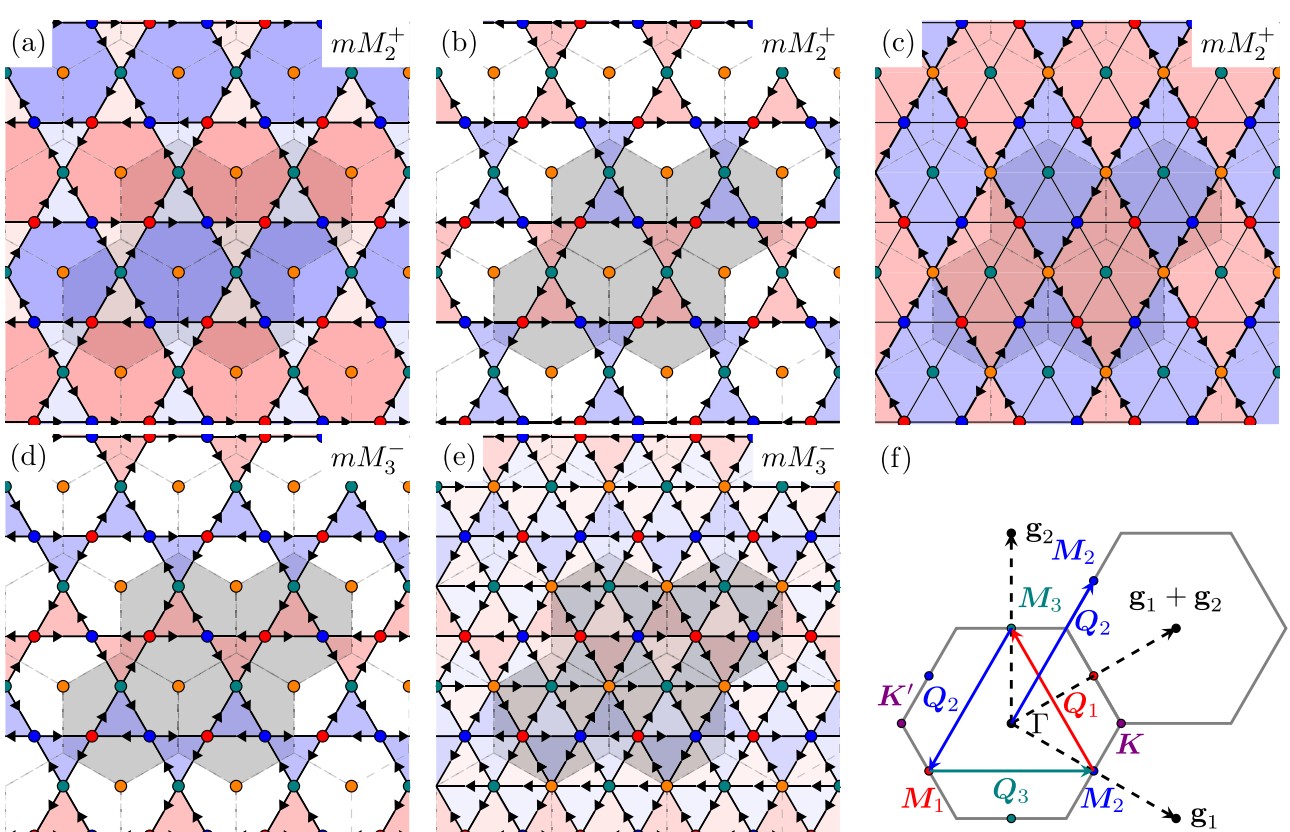

**Fig. 3 | Loop current patterns and Brillouin zone.** The 5 LC pattern possibilities that we study **a**–**e** and a sketch of the Brillouin zone (**f**). Each pattern comes in a set of 3, with ordering wave vectors $\mathbf{M}_{\ell=1, 2, 3}$. Only the representative with the ordering wave vector $\mathbf{M}_3$ is shown. The other two may be obtained via $C_{6z}$ rotations. The red, blue, and teal sites are V, and they form a kagome lattice. The orange sites are the planar Sb, and they form a triangular lattice and live in the centers of the hexagons formed by the kagome lattice. This reflects the same color scheme as in Fig. 1. The greyed-out region is the extended 2 × 2 unit cell that our LC patterns live in. The red and blue shadings denote whether the flux of a given plaquette is out-of-page or in-page, respectively. **a**–**c** LC patterns belong to the (even-parity) $mM_2^+$ irrep, while **d**–**e** belong to the (odd-parity) $mM_3^-$ irrep; see Supplementary, Section 1. **a**, **b**, **d** have only V-V currents, while **c**, **e** have in addition V-Sb currents. Note that the ordering wave vectors $\mathbf{Q}_\ell$, which connect different $\mathbf{M}$-points, are themselves $\mathbf{M}$-points (e.g., $\mathbf{Q}_3 = \mathbf{M}_2 - \mathbf{M}_1 = \mathbf{M}_3 = -\mathbf{M}_3$ up to a reciprocal lattice vector). The color scheme for these three $\mathbf{M}$-vectors (red, blue, teal) is chosen to coincide with the sublattice weight of the kagome sites (also red, blue, teal) at different $\mathbf{M}$-points in the Brillouin zone.

space group of this extended $2 \times 2$ unit cell. The character table and group operations of this reduced space group are detailed in the Supplementary Section 1. These LC patterns belong to three-dimensional irreps, with each pattern in the irrep possessing one of the three ordering vectors $\mathbf{Q}_\ell$, which transform into each other under a three-fold rotation. The analysis then yields three fermionic bilinears ($\ell = 1, 2, 3$)

$$\hat{J}_{\mathbf{q}}^{\ell} = \sum_{\mathbf{k}, \sigma} \hat{c}_{\mathbf{k}\sigma}^{\dagger} J^{\ell}(\mathbf{k}, \mathbf{k}+\mathbf{q}) \hat{c}_{\mathbf{k}+\mathbf{q}\sigma} \tag{1}$$

whose expectation values determine the three-component order parameter of the LC state. Here, $\hat{c}_{\mathbf{k}\sigma}$ is a 13-component spinor that annihilates an electron of spin $\sigma$ and momentum $\mathbf{k}$ in one of the 13 orbitals that is odd under the mirror symmetry $\sigma_h$. The explicit forms of the $13 \times 13$ matrices $J^{\ell}(\mathbf{k}, \mathbf{k}+\mathbf{q})$, which are determined by each specific LC pattern, are discussed in the Methods section. We remark that the group theoretic analysis does not fully constrain the loop current patterns, as they must also satisfy local charge conservation at every site[95,98] (Kirchhoff's law). The specific weights of current operators are shown in Table 1, and respect this local conservation law.

More important than the symmetry classification turns out to be the analysis of the current flow within each irrep. In particular, there are several LC states of identical transformation behavior (i.e., that transform under the same irrep), yet with very different microscopic pathways for the coherent circulation of charge. This is shown in Fig. 3 for LCs that transform under the irreducible representation $mM_2^+$ (the $m$ stands for time-reversal odd and the superscript $\pm$ for parity even/odd), which corresponds to Fig. 3a–c. Notice, in Fig. 3 we only show LC patterns with ordering wave vector $\mathbf{Q}_3 = (0, \frac{2\pi}{\sqrt{3}})$. For each panel, there are two additional patterns (shown in Supplementary Section 2) that are obtained by performing a six-fold rotation around the $z$-axis and which together comprise a three-dimensional space group irrep. As we shall see, the relative weights of the microscopic pathways depicted in Fig. 3a–c – which are all of the same symmetry – dictate the resulting SC state. Since no experimental constraint exists thus far that favors one pathway over the other, our phase diagrams will show the pairing state as a function of the relative weight between these LC pathways.

**Effective electron-electron interaction and pairing instabilities**

The interplay between LC order and superconductivity can be analyzed from various points of view. Previous studies have primarily focused on competing LC and SC instabilities[44,50] or simpler toy models[99–101]. Here, we study the superconductivity mediated by LC fluctuations from the disordered side of the phase diagram, i.e., before the collective LC modes have condensed (Fig. 1c). This is analogous to studies of SC mediated through the exchange of a nematic, spin magnetic, phononic, or other collective mode[95,98].

The effective low-energy electron-electron interaction that is mediated by fluctuating LCs is given by

$$\hat{H}_{\text{int}} = -\frac{g^2}{2N} \sum_{\mathbf{q}} \sum_{\ell, \ell'=1}^{3} [\mathcal{D}_{\text{LC}}(\mathbf{q})]_{\ell\ell'} \hat{J}_{-\mathbf{q}}^{\ell} \hat{J}_{\mathbf{q}}^{\ell'}, \tag{2}$$

where $N$ is the number of unit cells and $g$ is a coupling constant. We approach the modeling of the LC propagator $\mathcal{D}_{LC}(\mathbf{q})$ in two ways. In the first, we deduce a simple phenomenological form from symmetries and physical considerations. In the second, we use the random phase approximation (RPA) to calculate the propagator. As it will turn out, both predict the same pairing symmetries and qualitative behavior for the LC-mediated superconductivity.

To describe $M$-point LCs, the diagonal components of the LC propagator $[\mathcal{D}_{LC}(\mathbf{q})]_{\ell\ell}$ should be peaked at $\mathbf{Q}_\ell \cong \mathbf{M}_\ell$. Moreover, since we are approaching the problem from the disordered side, the LC propagator must obey the symmetries of the kagome lattice. The simplest

phenomenological ansatz consistent with these requirements is the following:

$$[\mathcal{D}_{\text{LC}}(\mathbf{q})]_{\ell\ell'} = \delta_{\ell\ell'}[r + (1-r)f(\mathbf{q} - \mathbf{M}_\ell)]^{-1}, \tag{3}$$

where $f(\mathbf{q}) = \frac{2}{3} - \frac{2}{9}(\cos(\mathbf{q} \cdot \mathbf{a}_1) + \cos(\mathbf{q} \cdot \mathbf{a}_2) + \cos(\mathbf{q} \cdot \mathbf{a}_3))$ with $\mathbf{a}_1 = (1, 0)$, $\mathbf{a}_2 = (\frac{1}{2}, \frac{\sqrt{3}}{2})$, and $\mathbf{a}_3 = \mathbf{a}_2 - \mathbf{a}_1$. Alternatively, we can explicitly compute the propagator of the loop currents within RPA, in which case it is given by

$$
\begin{aligned}
[\mathcal{D}_{\text{RPA}}^{-1}(\mathbf{q}, iq_0)]_{\ell\ell'} &= \delta_{\ell\ell'} \\
&+ \frac{1}{\beta NV} \sum_k \text{tr}[\mathcal{G}_0(k)J^{\ell}(\mathbf{k}, \mathbf{k}+\mathbf{q})\mathcal{G}_0(k+q)J^{\ell'}(\mathbf{k}+\mathbf{q}, \mathbf{k})].
\end{aligned} \tag{4}
$$

Here, $\mathcal{G}_0(k)$ is the ($13 \times 13$ matrix) Green function of the tight-binding model, and $J^\ell$ are the $13 \times 13$ matrices from Eq. (1). Due to imperfect nesting of the band structure, the RPA propagator will generally be peaked away from the $M$-points, and has structure elsewhere in the Brillouin zone which does not change the pairing state. We include a plot in the Supplementary Section 3 comparing the RPA propagator in Eq. (4) with the phenomenological one from Eq. (3).

This interaction can be projected onto the Cooper channel to give a gap equation. Since we are interested in the leading instability, at weak-coupling we may linearize this gap equation to obtain an eigenvalue problem:

$$\lambda \Delta_n(\mathbf{k}) = -\sum_{n'} \oint_{\text{FS}_{n'}} \frac{U_{nn'}^{s/t}(\mathbf{k}, \mathbf{k}')\Delta_{n'}(\mathbf{k}')dk'}{|\nabla_{\mathbf{k}'} \xi_{\mathbf{k}'n'}|(2\pi)^2}. \tag{5}$$

Here, $\Delta_n(\mathbf{k})$ is the singlet/triplet pairing amplitude in band $n$ with wave vector $\mathbf{k}$, $U_{nn'}^{s/t}(\mathbf{k}, \mathbf{k}')$ is the interaction in the singlet/triplet Cooper channel between points $\mathbf{k}, \mathbf{k}'$ in bands $n, n'$, respectively, and $\xi_{\mathbf{k}n}$ is the dispersion. The singlet and triplet channels are decoupled because the band Hamiltonian is inversion-symmetric. Since neither LC fluctuations nor the band Hamiltonian (which has no spin-orbit coupling) breaks the spin rotation symmetry, the triplet channel $\mathbf{d}$-vector may point in any direction, $\mathbf{d}_n(\mathbf{k}) = \Delta_n(\mathbf{k})\hat{\mathbf{n}}$. If all $\lambda < 0$, then there is no SC instability. We therefore search for the largest $\lambda > 0$ which corresponds to an SC instability with the largest $T_c \propto e^{-1/\lambda}$. When $U(\mathbf{k}, \mathbf{k}') < 0$ is attractive, then this cancels the leading minus sign in the gap equation, and $\Delta$ generically has the same sign at $\mathbf{k}$ and $\mathbf{k}'$. Conversely, if $U(\mathbf{k}, \mathbf{k}') > 0$ is repulsive, then the $\lambda > 0$ solutions favor a situation wherein $\Delta(\mathbf{k})$ and $\Delta(\mathbf{k}')$ have opposite signs.

As discussed in related contexts[95,102], under certain conditions $\lambda_{\text{max}}$ can diverge upon approaching a quantum-critical point. While our weak-coupling approach clearly breaks down in this regime, it is established that strong-coupling treatments can regularize this divergence, yielding the maximum in $T_c$ at the transition, as sketched in Fig. 1c, without altering the pairing symmetry identified in the weak-coupling analysis[103–107]. There is therefore good reason to believe that our weak-coupling theory – which enables us to address a problem involving a complex pairing interaction with a large number of orbitals per unit cell – can identify the leading pairing symmetry. We also note that the charge bond ordered (with potential loop current component) state in the kagome superconductors experiences a first-order transition[108] to a disordered state (at around 2 GPa in $CsV_3Sb_5$[34]), meaning that the mass never truly becomes zero at the transition.

For LC patterns that only have currents flowing between vanadium atoms (patterns a,b, d in Fig. 3), the dominant pairing instability (at the level of the linearized gap equation) is in the $E_{2g} = \{d_{x^2-y^2}, d_{xy}\}$ irrep. To see why this is the case, in Fig. 4a–c we have plotted the Fermi-surface-projected singlet-channel pairing interaction $U_s(\mathbf{k}, \mathbf{k}')$, which is the same interaction appearing in Eq. (5), for the V-V LC pattern of Fig. 3a. As one can see in Fig. 4(a), the interaction between the white point near $\mathbf{M}_3$ and the other $\mathbf{M}$-points is repulsive. The best way to

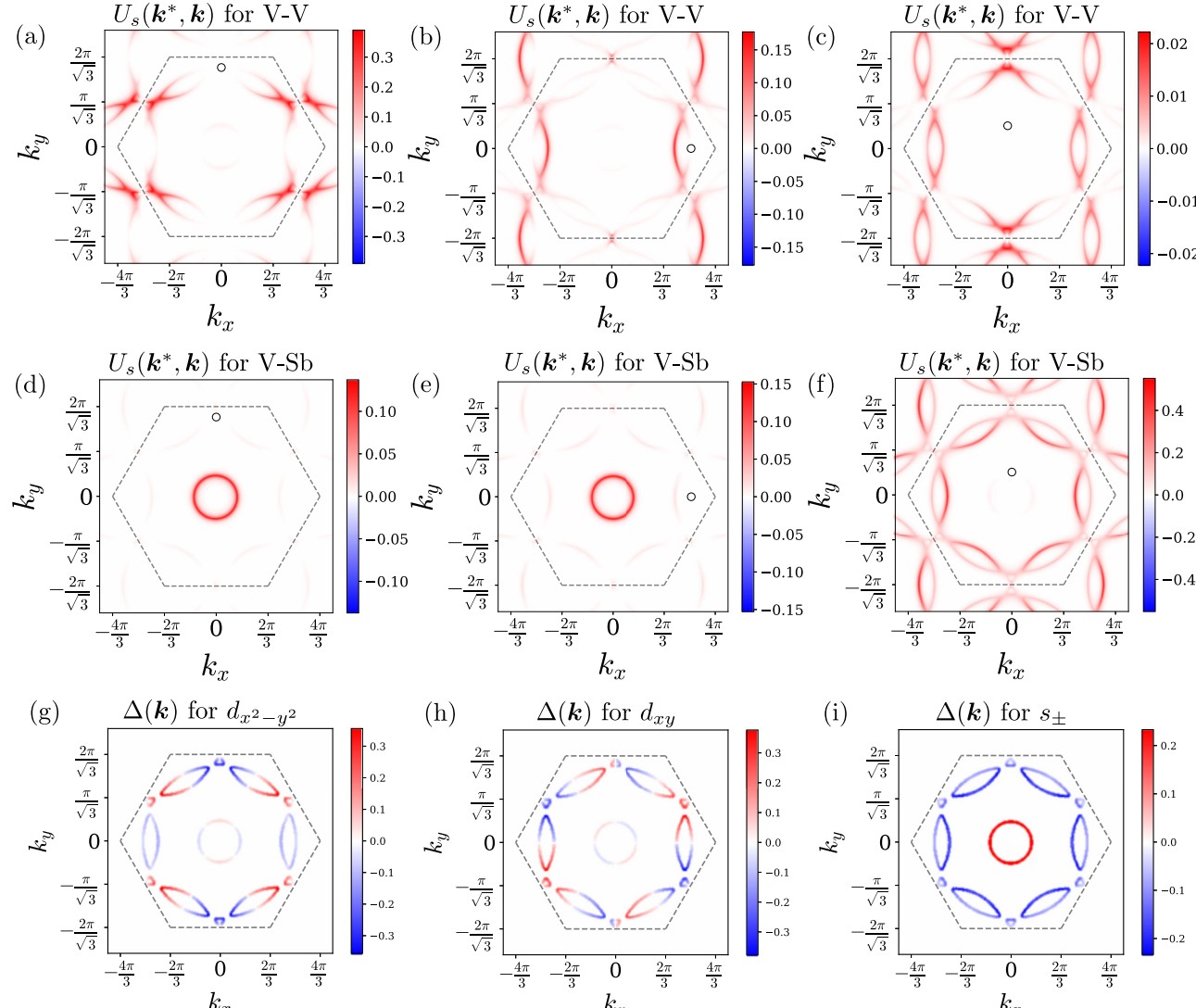

**Fig. 4 | Projected pairing kernels and gap functions.** The singlet-channel pairing interaction for two LC patterns **a**–**f** and examples of SC gap functions they result in (**g**)–(**i**). Under (**a**)–(**f**), one momentum of the interaction **k**˙ is fixed to the white point, which is on a Fermi surface, while the other momentum **k** is varied across the remaining Fermi surface area, with red color indicating the degree of repulsion. As mentioned in the discussion, the singlet interactions is always repulsive for interactions mediated by LCs. **a**–**c** show the interaction for the V-V $mM_2^+$ pattern of

Fig. 3a. **d**–**f** correspond to the V-Sb $mM_2^+$ patterns of Fig. 3c. In (**g**) we show the basis gap function for $d_{x^2-y^2}$ pairing, in (**h**) for $d_{xy}$, and in (**i**) for $s^\pm$. In hexagonal systems, $d_{x^2-y^2}$ and $d_{xy}$ belong to the same two-dimensional irrep, namely $E_{2g}$. All plots in this figure were generated for mass $r = 0.3$, and used the phenomenological propagator from Eq. (3). Using the RPA propagator yields indistinguishable results for the pairing states.

achieve pairing from this repulsive interaction is through $d_{x^2-y^2}$ or $d_{xy}$ pairing since the gap function changes sign between the **M**-points. The corresponding SC gap functions are displayed in Fig. 4g, h. After introducing non-linear corrections to the linearized gap equation, we find that the time-reversal symmetry-breaking state $d_{x^2-y^2} + id_{xy}$ is favored. The $d_{x^2-y^2} + id_{xy}$ state is fully gapped, in contrast to $d_{x^2-y^2}$ or $d_{xy}$, each of which has nodal lines. Importantly, because there is almost no coupling between the outer sheets of the Fermi surface and the planar Sb pocket near the **Γ**-point, the SC gap function can be negligibly small on the inner **Γ**-pocket, with a sign that has no reason to be opposite to the outer Fermi surface sheets. In turn, this implies that $d_{x^2-y^2} + id_{xy}$, or more simply $d + id$, pairing is insensitive to the presence of the **Γ**-pocket. The triplet-channel interactions are, for completeness, shown in the Supplementary Section 4.

On the other hand, if we have currents flowing between V and Sb atoms (the pattern in Fig. 3c), there exists a strong repulsion between the circular Fermi surface around **Γ** and the parts of the Fermi surface near the **M**-points, as shown in Fig. 4d–f. This indicates that the gap on

the outer sheets of the Fermi surface must have the opposite sign compared to the Sb sheet near **Γ**. Consequently, the leading solution for this case has $s^\pm$ symmetry, as indicated by the gap structure in Fig. 4i. The pattern of Fig. 3e has both V-V and V-Sb loop currents coexisting, so whether $d + id$ or $s^\pm$ is favored depends on the specific ratio of energy scales between V-V interactions and V-Sb interactions. In the case we have chosen, this pattern yields $s^\pm$ pairing. The fact that fluctuating V-V LC patterns favor $d + id$ pairing, while fluctuating V-Sb LC patterns favor $s^\pm$ pairing, is the primary result of this paper.

Having studied the two cases separately, we now address what happens if we consider a generic superposition of V-V and V-Sb LC patterns. In Fig. 3, patterns (a)–(c) all belong to the same irrep $mM_2^+$ of the reduced space group. This means that they can be added together with arbitrary coefficients to form a new current pattern belonging to the same irrep. We therefore study how the leading SC state depends on the normalized combination $\widehat{J}^\ell = \sin\theta\cos\phi\widehat{J}^\ell_{(a)} + \sin\theta\sin\phi\widehat{J}^\ell_{(b)} + \cos\theta\widehat{J}^\ell_{(c)}$. The subscripts (a)–(c) refer to the patterns in Fig. 3a–c. The corresponding phase diagram is illustrated in Fig. 5a. Evidently, when $\theta$

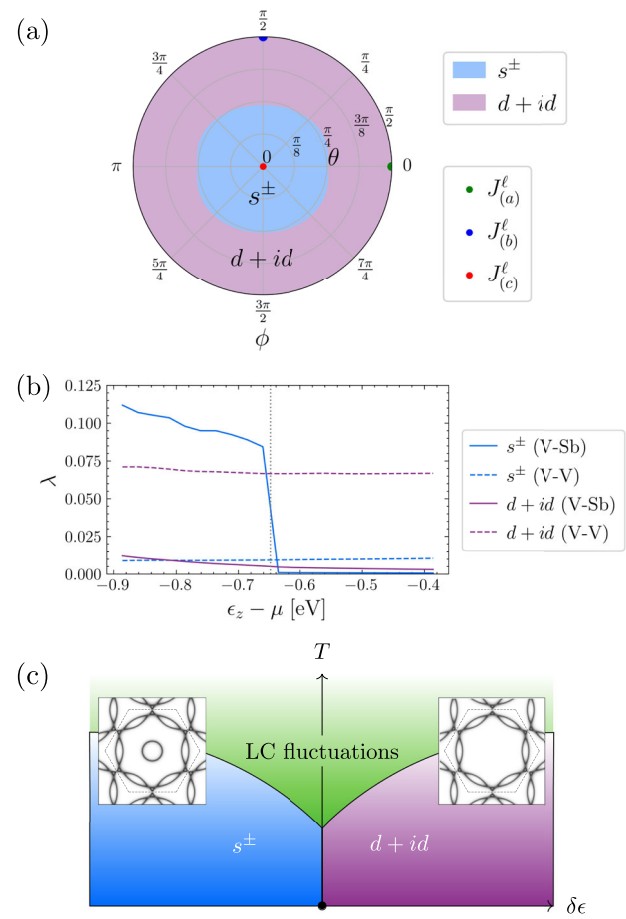

**Fig. 5 | Phase diagram of pairing symmetries under various dominant LC pathways. a** A polar plot showing how the dominant pairing instability depends on the LC pattern, as parametrized by $\widehat{J}^{\ell} = \sin\theta\cos\phi\widehat{J}_{(a)}^{\ell} + \sin\theta\sin\phi\widehat{J}_{(b)}^{\ell} + \cos\theta\widehat{J}_{(c)}^{\ell}$. The variables $\widehat{J}_{(a)}^{\ell}, \widehat{J}_{(b)}^{\ell}, \widehat{J}_{(c)}^{\ell}$ refer to the current patterns of Fig. 3a, b, c, respectively. The plot is periodic in $\phi$. $\theta \in [\frac{\pi}{2}, \pi]$ is not shown because $\widehat{J}^{\ell}$ and $-\widehat{J}^{\ell}$ both give the same interaction. When $\theta$ (radial variable) is small, the currents flow predominantly between V-Sb, driving $s^{\pm}$ pairing. **b** The phase diagram as a function of $\epsilon_z$, which is the tuning parameter for the $\Gamma$-point pocket, for $\widehat{J}_{(a)}$ (dashed lines) and $\widehat{J}_{(c)}$ (solid lines) exchange. Since $\widehat{J}_{(a)}$ is a current pattern flowing only between V-V, the $\Gamma$-pocket plays essentially no role, so its removal (vertical dotted line) does not affect the leading pairing eigenvalue. In contrast, because $\widehat{J}_{(c)}$ flows only between V-Sb, which strongly couples the $\Gamma$-pocket and outer Fermi surface, the removal of the $\Gamma$-pocket destroys the $s^{\pm}$ superconductivity; only weak interactions between the M-points on the outer Fermi surface remain, yielding $d + id$ pairing. **c** A schematic phase diagram illustrating the quantitative result of (**b**), wherein the removal of the $\Gamma$ pocket at the Lifshitz transition destroys the $s^{\pm}$ state. **a** and **b** were calculated using the RPA LC propagator from Eq. (4).

is small, we have predominantly $s^{\pm}$ pairing, as expected given that the current pattern is predominantly V-Sb of the type shown in Fig. 3c, with interactions depicted in Fig. 4d–f. If, instead, $\theta$ is close to $\frac{\pi}{2}$, then the LC pattern is dominated by currents flowing between V-V sites, and consequently the interaction closely resembles those of Fig. 4a–c. This leads to $d + id$ pairing, as expected. We include in the Supplementary Section 4 the plots of the projected pairing interaction for all loop current patterns, in addition to the ones shown in Fig. 4.

The next issue to address is the fate of the SC solution arising from the V-Sb LC pattern (Fig. 3c) when the $\Gamma$-pocket is removed. Experimentally, in CsV$_3$Sb$_5$ the $\Gamma$-pocket undergoes a Lifshitz transition (i.e., it moves above the Fermi level) at approximately 7.5 GPa, which is around the same point at which the superconductivity disappears[79]. In our tight-binding model, we can tune the on-site potential $\epsilon_z$ of the

planar Sb $p_z$ orbital to move this circular pocket above the Fermi energy. Upon doing this, we find that the $s^{\pm}$ superconductivity is strongly suppressed at the Lifshitz transition (signaled by the vertical dotted line in Fig. 5b), and that the dominant eigenvalue once again falls into the $E_{2g}$ irrep. The leading eigenvalue and its symmetry classification as a function of $\epsilon_z$ is depicted in Fig. 5(b).

## Discussion

In our analysis of superconductivity due to LC fluctuations, we have considered a wide array of different LC patterns with translation symmetry-breaking ($2 \times 2$ unit cell) in line with scanning tunneling microscopy and X-ray scattering experiments. We considered states of even and odd parity, with and without currents flowing to the planar Sb atoms. The main result of our work is that current patterns flow only between V yield $d + id$ SC, yet current patterns also flowing to planar Sb yield $s^{\pm}$ SC, regardless of the parity of the LC state. These results can be understood as follows.

It is a general feature of boson-mediated pairing that a time-reversal even boson mediates an attractive (in the singlet Cooper channel) interaction, whereas a time-reversal odd boson mediates a repulsive interaction[98]. Common examples of the former are attraction mediated by phonons or nematic fluctuations. On the other hand, spin fluctuations and the very LC fluctuations described in this paper mediate repulsive interactions. It should therefore come as no surprise that the singlet-channel interactions in Fig. 4a–f are nonnegative (i.e., repulsive) everywhere in the Brillouin zone. As long as the repulsive interaction is between distinct points on the Fermi surface (i.e., for finite momentum exchange), unconventional SC solutions are typical[109]. The fact that the LC propagator peaks at (or nearby) M-points means that the repulsion is large between the different M-points, highlighting the effectiveness of translation symmetry-breaking loop currents in driving superconductivity. Furthermore, the pairing eigenvalue $\lambda$ is enhanced as the boson mass $r \to 0$, making the interaction most efficient near a LC quantum-critical point.

The V-V patterns, regardless of whether they have even or odd parity, consist of electrons on different V sublattices interacting with (and, furthermore, repelling) one another. In momentum space, such an interaction couples electrons at different M-points. This incentivizes sign-changing of the gap function between different M-points, and is the origin of the $d_{x^2-y^2} + id_{xy}$ pairing. On the other hand, in a V-Sb pattern, electrons from V sites repel the electrons from the in-plane Sb. In momentum space, this leads to a strong repulsion between electrons near M-points and the circular Fermi surface around the $\Gamma$-point. Such a strong repulsion makes the gap function change its sign between the inner $\Gamma$-pocket and the M-point electrons, giving $s^{\pm}$ pairing. It is interesting to note that pairing states of different symmetries are favored by LC patterns with the same symmetry, promoted by different orbital compositions. This is reminiscent of the physics of iron-based superconductors, for which spin fluctuations can favor either $s^{\pm}$ or $d_{x^2-y^2}$ wave pairing[110].

How does our theory of LC-induced unconventional pairing compare with experimental observations of the superconducting gap structure in AV$_3$Sb$_5$? Both candidate pairing states − $s^{\pm}$ and chiral $d + id$ − are fully gapped (apart from possible accidental nodes), consistent with experimental observations[61,111,112]. Moreover, recent quasiparticle interference measurements[67] indicate a large SC gap on the $\Gamma$-pocket Fermi surface, which favors the $s^{\pm}$ pairing candidate. The disappearance of superconductivity in CsV$_3$Sb$_5$ around 7.5 GPa[79,84], coinciding with the Lifshitz transition that eliminates the $\Gamma$ pocket, further underscores the importance of planar Sb states in mediating superconductivity. Remarkably, a second SC dome appears at approximately 16.5 GPa[79,84], which our theory naturally interprets as a transition from the $s^{\pm}$ state to the $d + id$ state, as shown in Fig. 5c. Superconductivity near ambient pressure being of the $s^{\pm}$ type is also consistent with electron irradiation experiments[113], which show that

the SC state remains robust in the presence of charge impurities. This is expected, as $s^{\pm}$ pairing is known to be more resilient to impurity scattering than chiral $d + id$ pairing[114]. Finally, $\mu$SR experiments at low pressure ($P < 7.5$ GPa) observe a change in the muon relaxation rate below $T_c$, including in systems lacking CDW order[60,61]. This has been interpreted as evidence for time-reversal symmetry-breaking, which would naïvely suggest $d + id$ pairing in this regime. However, recent insights from the case of $Sr_2RuO_4$ show that similar $\mu$SR signatures can arise from closely competing pairing states, with time-reversal symmetry being locally broken near strain inhomogeneities and dislocations[115–117]. Our theory provides a natural explanation for such near-degenerate superconducting states. We therefore expect that local $E_{2g}$ strain fields may stabilize time-reversal symmetry-breaking admixtures of the dominant $s^{\pm}$ and subleading $d + id$ pairing components. Overall, the chiral $d + id$ state remains a viable candidate in parts of the phase diagram (Fig. 5). The pressure-induced transition between $s^{\pm}$ and $d + id$ pairing states comes naturally as an explanation for the observations of ref. 79,84 within our framework.

One aspect that we have neglected is the large hexagonal-shaped Fermi surface that passes close to the **M**-points and that is due to orbitals which are even under $\sigma_h: z \mapsto -z$. This part of the Fermi surface may be identified in Fig. 2a (but is, of course, absent in Fig. 2b). To understand the impact of these degrees of freedom, we observe that, because our interactions are generated purely through fluctuating LC modes made up of $\sigma_h$-odd LC patterns, they only couple $\sigma_h$-odd orbitals to other $\sigma_h$-odd orbitals. However, even at $k_z = 0$, the $\sigma_h$-even orbitals should couple to $\sigma_h$-odd orbitals via additional interactions, such as the Coulomb interaction. The gap opened up by either $d + id$ or $s^{\pm}$ superconductivity on the $\sigma_h$-odd bands is therefore expected to induce a gap on the $\sigma_h$-even bands as well.

In conclusion, fluctuating translation symmetry-breaking LC patterns are an effective pairing glue for superconductivity. The two candidate states we find are $d + id$ and $s^{\pm}$, both of which are fully gapped states. In particular, to obtain $s^{\pm}$ superconductivity, little explored LC patterns that traverse between V and Sb atoms must be included in the analysis. Our results for the $s^{\pm}$ state are consistent with the crucial role of planar Sb states[73,77] for superconductivity below 7.5 GPa[79,84], whereas our results for the $d + id$ state also present a potential pairing mechanism for superconductivity beyond 16.5 GPa[79,84], at which point no **Γ**-pocket remains[73,77]. The pairing sym-

metry of the superconductivity depends crucially on the microscopic pathway of the loop current, highlighting the significance of orbital-selective LC interactions, and their implications for couplings on the Fermi surface.

## Methods

### Effective electron-electron interaction

Here, we present the explicit expressions for the coupling between the electrons and the LC boson corresponding to the several LC patterns drawn in Fig. 3 of the main text. Consider a hexagon as shown in Fig. 6. The index $n$ labels the unit cell that the planar Sb site lives in. The LC operator on a single hexagon has two different forms for the two cases of V-V or V-Sb patterns, respectively:

$$\widehat{J}_n = \sum_{c=1}^{6} \alpha_c \widehat{j}_c, \quad \text{V} - \text{V pattern}, \tag{6}$$

$$\widehat{J}_n = \sum_{c=1}^{6} \beta_c \widehat{j}_c, \quad \text{V} - \text{Sb pattern}. \tag{7}$$

The meanings of the coefficients $\alpha_c$ and $\beta_c$ are indicated in Fig. 6. The subscript $c$ labels the different bonds and $\widehat{j}_c$ is the current operator on the specified bond; see Fig. 6. By specifying the $\alpha_c$, $\beta_c$ coefficients,

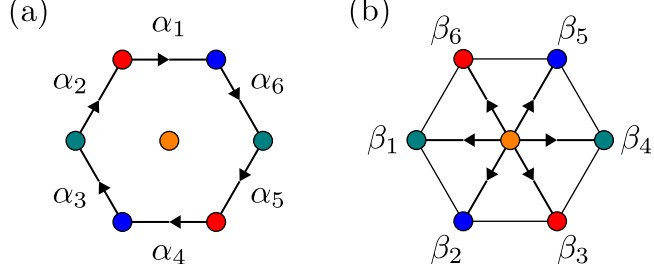

**Fig. 6 | Labels for coefficients of the current operator within a single hexagon.** In both cases, we take the convention that setting the coefficient positive yields the indicated direction of current. **a** The case of V-V currents. **b** The case of V-Sb currents.

**Table 1 | Current patterns within a unit cell**

| Pattern | Wave vector(s) | $(\alpha_1, \alpha_2, \alpha_3, \alpha_4, \alpha_5, \alpha_6)$ | $(\beta_1, \beta_2, \beta_3, \beta_4, \beta_5, \beta_6)$ |
|---|---|---|---|
| (a) | $\mathbf{M}_1, \mathbf{M}_2, \mathbf{M}_3$ | $\frac{1}{\sqrt{6}}(1, 1, 1, 1, 1, 1)$ | $(0, 0, 0, 0, 0, 0)$ |
| | $\mathbf{M}_1$ | $\frac{1}{\sqrt{12}}(1, 1, -2, 1, 1, -2)$ | $(0, 0, 0, 0, 0, 0)$ |
| (b) | $\mathbf{M}_2$ | $\frac{1}{\sqrt{12}}(1, -2, 1, 1, -2, 1)$ | $(0, 0, 0, 0, 0, 0)$ |
| | $\mathbf{M}_3$ | $\frac{1}{\sqrt{12}}(-2, 1, 1, -2, 1, 1)$ | $(0, 0, 0, 0, 0, 0)$ |
| | $\mathbf{M}_1$ | $(0, 0, 0, 0, 0, 0)$ | $\frac{1}{2}(-1, 1, 0, -1, 1, 0)$ |
| (c) | $\mathbf{M}_2$ | $(0, 0, 0, 0, 0, 0)$ | $\frac{1}{2}(1, 0, -1, 1, 0, -1)$ |
| | $\mathbf{M}_3$ | $(0, 0, 0, 0, 0, 0)$ | $\frac{1}{2}(0, -1, 1, 0, -1, 1)$ |
| | $\mathbf{M}_1$ | $(0.358, -0.358, -0.494, -0.358, 0.358, 0.494)$ | $(0, 0, 0, 0, 0, 0)$ |
| (d) | $\mathbf{M}_2$ | $(-0.358, -0.494, -0.358, 0.358, 0.494, 0.358)$ | $(0, 0, 0, 0, 0, 0)$ |
| | $\mathbf{M}_3$ | $(0.494, 0.358, -0.358, -0.494, -0.358, 0.358)$ | $(0, 0, 0, 0, 0, 0)$ |
| | $\mathbf{M}_1$ | $-\frac{0.77535}{\sqrt{6}}(1, -1, 1, -1, 1 - 1)$ | $\frac{0.63153}{\sqrt{6}}(-1, 1, 1, 1, -1, -1)$ |
| (e) | $\mathbf{M}_2$ | $-\frac{0.77535}{\sqrt{6}}(-1, 1, -1, 1, -1, 1)$ | $\frac{0.63153}{\sqrt{6}}(1, 1, 1, -1, -1, -1)$ |
| | $\mathbf{M}_3$ | $-\frac{0.77535}{\sqrt{6}}(-1, 1, -1, 1, -1, 1$ | $\frac{0.63153}{\sqrt{6}}(-1, -1, 1, 1, 1, -1)$ |

The pattern labels reference the patterns in Fig. 3 of the main text. In the main text, only the patterns with ordering wave vector $\mathbf{Q}_3$ are shown. The cases **a**, **b**, **c** have very simple coefficients because the Kirchhoff law is enforced for symmetry reasons. In contrast, the cases (**d**), (**e**) require fine-tuning of the parameters.

one can construct the current operator $\hat{j}_n$ and hence $\hat{j}_{\mathbf{q}}$. The coefficients for these patterns are listed in Table 1.

By calculating the Fourier transform $\hat{j}_{\mathbf{q}} = \sum_n e^{-i\mathbf{R}_n \cdot \mathbf{q}} \hat{j}_n$, these couplings lead to the following interaction term

$$S_{\text{LC-el}} = -\frac{g}{\beta N} \sum_{iq_0, \mathbf{q}} \sum_{\ell=1}^{3} J_q^\ell \Phi_q^\ell, \tag{8}$$

where the sum over $\ell$ goes over the three different **M**-points. Here, the fermionic bilinears couple to real, time-reversal odd bosons, which we call $\Phi$, representing the collective LC modes. There are three bosons, one for each **M**-point. The kinetic term for the bosons consists of the inverse propagator:

$$S_\Phi = \frac{1}{2\beta N} \sum_q \sum_{\ell, \ell'=1}^{3} [\mathcal{D}_{\text{LC}}^{-1}(\mathbf{q})]_{\ell\ell'} \Phi_{-q}^\ell \Phi_q^{\ell'}, \tag{9}$$

The loop current propagator $[\mathcal{D}_{LC}(\mathbf{q})]_{\ell\ell'}$ has its maximum at the ordering wave vector of the boson.

### Integrating out the LC boson
We integrate out the LC boson by performing the following Gaussian integral:

$$\int \exp\left(-\frac{1}{\beta N} \sum_q \left[\frac{1}{2}[\mathcal{D}_{\text{LC}}^{-1}(\mathbf{q})]|\Phi_q|^2 - g\Phi_q J_{-q}\right]\right)$$
$$\mathcal{D}\Phi = \exp\left(\frac{g^2}{2\beta N} \sum_q \mathcal{D}_{\text{LC}}(\mathbf{q})J_{-q}J_q\right). \tag{10}$$

This generates the following interaction which is mediated by LC fluctuations:

$$\hat{H}_{int} = -\frac{g^2}{2N} \sum_{\mathbf{q}} \sum_{\ell, \ell'=1}^{3} [\mathcal{D}_{\text{LC}}(\mathbf{q})]_{\ell\ell'} : \hat{J}_{-\mathbf{q}}^\ell \hat{J}_{\mathbf{q}}^{\ell'} :$$
$$= \frac{1}{2N} \sum_{\substack{\mathbf{k}_1 \mathbf{k}_2 \mathbf{k}_2' \mathbf{k}_1' \\ abb'a' \\ \sigma_1 \sigma_1' \sigma_2' \sigma_2}} V_{abb'a'}^{\sigma_1 \sigma_2 \sigma_2' \sigma_1'}(\mathbf{k}_1, \mathbf{k}_2, \mathbf{k}_2', \mathbf{k}_1') \hat{c}_{\mathbf{k}_1 a \sigma_1}^\dagger \hat{c}_{\mathbf{k}_2 b \sigma_2}^\dagger \hat{c}_{\mathbf{k}_2', b', \sigma_2'} \hat{c}_{\mathbf{k}_1', a', \sigma_1'}, \tag{11}$$

whereby now we explicitly indicate the sublattice/orbital indices $a, b, a', b'$ for the fermions. The interaction vertex is then defined according to

$$V_{abb'a'}^{\sigma_1 \sigma_2 \sigma_2' \sigma_1'}(\mathbf{k}_1, \mathbf{k}_2, \mathbf{k}_2', \mathbf{k}_1') = -g^2 \delta_{\mathbf{k}_1 + \mathbf{k}_2, \mathbf{k}_1' + \mathbf{k}_2'} \delta_{\sigma_1 \sigma_1'} \delta_{\sigma_2 \sigma_2'}$$
$$\sum_{\ell=1}^{3} [\mathcal{D}_{\text{LC}}(\mathbf{k}_1 - \mathbf{k}_1')]_{\ell\ell'} J_{aa'}^\ell(\mathbf{k}_1, \mathbf{k}_1') J_{bb'}^{\ell'}(\mathbf{k}_2, \mathbf{k}_2'). \tag{12}$$

Next, we explicitly antisymmetrize the interaction (of course, fermion statistics enforces this, but it is nonetheless useful to ensure the matrix elements also obey antisymmetrization)

$$U_{abb'a'}^{\sigma_1 \sigma_2 \sigma_2' \sigma_1'}(\mathbf{k}_1, \mathbf{k}_2, \mathbf{k}_2', \mathbf{k}_1') = V_{abb'a'}^{\sigma_1 \sigma_2 \sigma_2' \sigma_1'}(\mathbf{k}_1, \mathbf{k}_2, \mathbf{k}_2', \mathbf{k}_1') - V_{bab'a'}^{\sigma_2 \sigma_1 \sigma_2' \sigma_1'}(\mathbf{k}_2, \mathbf{k}_1, \mathbf{k}_2', \mathbf{k}_1') \tag{13}$$

so that the interaction is given by

$$\hat{H}_{int} = \frac{1}{4N} \sum_{\substack{\mathbf{k}_1 \mathbf{k}_2 \mathbf{k}_2' \mathbf{k}_1' \\ abb'a' \\ \sigma_1 \sigma_1' \sigma_2' \sigma_2}} U_{abb'a'}^{\sigma_1 \sigma_2 \sigma_2' \sigma_1'}(\mathbf{k}_1, \mathbf{k}_2, \mathbf{k}_2', \mathbf{k}_1') \hat{c}_{\mathbf{k}_1 a \sigma_1}^\dagger \hat{c}_{\mathbf{k}_2 b \sigma_2}^\dagger \hat{c}_{\mathbf{k}_2', b', \sigma_2'} \hat{c}_{\mathbf{k}_1', a', \sigma_1'}. \tag{14}$$

### Linearized gap equation
We may now project onto the Cooper channel by requiring that $\mathbf{k} := \mathbf{k}_1 = -\mathbf{k}_2$ and $\mathbf{k}' := -\mathbf{k}_2' = \mathbf{k}_1'$. This yields a Cooper channel interaction $U_{abb'a'}^{\sigma_1 \sigma_2 \sigma_2' \sigma_1'}(\mathbf{k}, -\mathbf{k}, -\mathbf{k}', \mathbf{k}')$. The corresponding BCS-type Hamiltonian is

$$\hat{H} = \sum_{\mathbf{k}ab\sigma} \hat{c}_{\mathbf{k}a\sigma}^\dagger [H_0(\mathbf{k})]_{ab} \hat{c}_{\mathbf{k}b\sigma} + \frac{1}{4N} \sum_{\substack{\mathbf{k}\mathbf{k}' abb'a' \\ \sigma_1 \sigma_1' \sigma_2' \sigma_2}} U_{abb'a'}^{\sigma_1 \sigma_2 \sigma_2' \sigma_1'}$$
$$(\mathbf{k}, -\mathbf{k}, -\mathbf{k}', \mathbf{k}') \hat{c}_{\mathbf{k}a\sigma_1}^\dagger \hat{c}_{-\mathbf{k}, b\sigma_2}^\dagger \hat{c}_{-\mathbf{k}', b', \sigma_2'} \hat{c}_{\mathbf{k}', a', \sigma_1'}. \tag{15}$$

By defining the gap function as

$$\Delta_{ab\sigma_1\sigma_2}(\mathbf{k}) = -\frac{1}{2N} \sum_{\mathbf{k}' b' a' \sigma_2' \sigma_1'} U_{abb'a'}^{\sigma_1 \sigma_2 \sigma_2' \sigma_1'}(\mathbf{k}, -\mathbf{k}, -\mathbf{k}', \mathbf{k}') \langle \hat{c}_{-\mathbf{k}' b' \sigma_2'} \hat{c}_{\mathbf{k}' a' \sigma_1'} \rangle, \tag{16}$$

we can perform the standard construction of the linearized gap equation[98] to obtain

$$\frac{\Delta_{ab\sigma_1\sigma_2}(\mathbf{k})}{\log(\beta\hbar\omega_c 2e^\gamma/\pi)} = -\frac{1}{2} \sum_{nb'a'} \oint_{\xi_n(\mathbf{k}')=0} U_{abb'a'}^{\sigma_1 \sigma_2 \sigma_2' \sigma_1'}(\mathbf{k}, -\mathbf{k}, -\mathbf{k}', \mathbf{k}')[T(-\mathbf{k}')]_{b'n}[T(\mathbf{k}')]_{a'n}$$
$$\times \sum_{cd\sigma_2'\sigma_1'} [T^\dagger(\mathbf{k}')]_{nc}[T^\dagger(-\mathbf{k}')]_{nd}[\Delta(\mathbf{k}')]_{cd\sigma_1'\sigma_2'} \frac{d\mathbf{k}'}{|\nabla_{\mathbf{k}'} \xi_n(\mathbf{k}')|(2\pi)^2}. \tag{17}$$

Here, the matrix $T$ is the change of basis matrix between orbitals and bands, defined through $[H_0(\mathbf{k})]_{ab} = \sum_n [T(\mathbf{k})]_{an} \xi_n(\mathbf{k}) [T^\dagger(\mathbf{k})]_{nb}$. Everything can appropriately be transformed to the band basis (here, $n$ labels bands, whereas $a, b, c, d$ label orbitals). The above linearized gap equation can be recast in the band basis, and then split into singlet and triplet components, yielding the final form of the gap equation found in the main text:

$$\lambda \Delta^{s/t}(\mathbf{k}) = -\oint_{\text{FS}} U_{s/t}(\mathbf{k}, \mathbf{k}') \Delta^{s/t}(\mathbf{k}') \frac{d\mathbf{k}'}{|\nabla\xi(\mathbf{k}')|(2\pi)^2} \tag{18}$$

Here, $U_{s/t}(\mathbf{k}, \mathbf{k}')$ is the interaction in the singlet/triplet channel. It is related to the earlier matrix elements of the interaction in the following way:

$$U^{AB}(\mathbf{k}_m, \mathbf{k}_n) = \sum_{\substack{\sigma_1 \sigma_2 \sigma_2' \sigma_1' \\ abb'a'}} [\sigma^A \sigma^y]_{\sigma_2 \sigma_1}$$
$$\frac{[T^\dagger(\mathbf{k})]_{ma}[T^\dagger(-\mathbf{k})]_{mb} U_{abb'a'}^{\sigma_1 \sigma_2 \sigma_2' \sigma_1'}(\mathbf{k}, -\mathbf{k}, -\mathbf{k}', \mathbf{k}')[T(-\mathbf{k}')]_{b'n}[T(\mathbf{k}')]_{a'n}}{4} [\sigma^y \sigma^B]_{\sigma_1' \sigma_2'} \tag{19}$$

This matrix element is proportional to $\delta_{AB}$ because of SU(2) spin invariance, and $m, n$ are actually just set by the point on the Fermi surface since for any given **k**-point on the Fermi surface, only one band crosses it. Thus $U^s$ corresponds to $A = B = 0$, and $U^t$ to $A = B = 1, 2, 3$. This is the interaction that is plotted in Fig. 4 of the main text.

### Data availability
All data required to reach the conclusions in the paper are present in the main text, supplementary, or cited references.

### Code availability
Code used in this project was deposited in CodeOcean: https://doi.org/10.24433/CO.9293789.v1

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

## Acknowledgements

We are grateful to Ronny Thomale, Steven Kivelson, Harshit Agarwal, Stuart Brown, Hans-Joachim Elmers, Olena Fedchenko, Mark Fischer, Zurab Guguchia, Amir A. Haghighirad, Matthieu Le Tacon, Titus Neupert, Eduardo H. da Silva Neto, and Roser Valentí for stimulating discussions. D.J.S. and J.S. disclose support for the research of this work from the German Research Foundation (DFG) through CRC TRR 288 "Elasto-Q-Mat," project A07. J.S. discloses support for the research of this work from the Simons Foundation Collaboration on New Frontiers in Super-conductivity (Grant SFI-MPS-NFS-00006741-03). Y.B.K. discloses support for the research of this work from the Natural Science and Engineering Research Council of Canada (NSERC) Discovery Grant no. RGPIN-2023-03296 and the Center for Quantum Materials at the University of Toronto. R.M.F. discloses support for the research of this work from the Air Force Office of Scientific Research under Award No. FA9550-21-1-0423, and a Mercator Fellowship from the German Research Foundation (DFG) through CRC TRR 288, 422213477 "Elasto-Q-Mat."

## Author contributions

J.S., R.M.F., and Y.B.K. conceived the project. Tight-binding and group theory calculations were performed by D.J.S., gap equation calculations by G.P., and RPA calculations by A.M. The manuscript was written by D.J.S., J.S., and G.P., with input by R.M.F and Y.B.K. All authors contributed to the interpretation and discussion.

## Funding

## Competing interests

The authors declare no competing interests.
