## [Transparent Peer Review file · Nature Communications]

Superconductivity in kagome metals due to soft loop-current fluctuations

Corresponding Author: Dr Daniel Schultz

Version 0:

Reviewer comments:

Reviewer #1

(Remarks to the Author)

In this manuscript, "Superconductivity in kagome metals due to soft loop-current fluctuations", the authors illustrate that soft fluctuations of translation symmetry-breaking loop currents (LCs) provide a mechanism for unconventional superconductivity (SC) in kagome metals AV_3Sb_5 . They argue that the fluctuations of LCs with different pathways lead to two distinct pairing states, with V-V LCs favor chiral $d+id$ state and V-Sb LCs favor s_{\pm} state. The theory provides a natural explanation for the experimental phase diagram of CsV_3Sb_5 under pressure where a Lifshitz transition takes place at ~ 7.5 GPa, beyond which the Sb-dominant Γ -pocket is removed.

This work presents a comprehensive theoretical framework for the unconventional superconductivity in kagome metals, and emphasizes the crucial role of multi-orbital physics, particularly the role of Sb 5p orbitals. The approach is standard and well-established in the field, and the statements and arguments made in the manuscript are valid and supported by reasonable experimental evidence. It is a promising contribution to the field of kagome superconductivity and would stimulate significant discussion and further research. I would recommend publication in Nature Communications after the authors properly address the following concern.

1) The LC susceptibility employed in Eq. (2) is phenomenological. While this manuscript emphasizes the crucial role of multi-orbital physics in capturing the correct low-energy physics and pairing instabilities, the multi-orbital physics is not implemented in the phenomenological form of the LC susceptibility. An investigation with LC susceptibility computed from the underlying microscopic model would further strengthen the theory.

2) Are the findings sensitive to the detail electronic structure, Fermi surface topology in particular, of the multi-orbital model? According to DFT calculations [see for example Fig. 2a in PRB 108, L060503 (2023) and Fig. 1b in PRB 111, 235114 (2025)], the two hexagonal Fermi surfaces whose vertices point to the M-point in the BZ are from σ_h -odd orbitals, while the hexagonal Fermi surface whose vertices point to the K-point (the rounded Fermi surface in this work) is due to σ -even orbitals. This seems to be little bit different from the Fermi surface topology and orbital content in the multi-orbital model adopted in this work. Since the vertices carry large amount of density of states, could this difference lead to qualitative different results?

3) In Fig. 3, the authors sketched the representative LC patterns with ordering wave vector M_3 . It might be helpful for the readers if the authors can show, somewhere in the supplemental, the combined LC patterns from all three ordering wave vectors, and make connections to the V-V LC patterns proposed in previous works, i.e., the pattern proposed in Science Bulletin 66, 1384 (2021) and the four patterns obtained by mean-field calculations in PRB 107, 045127 (2023).

4) It is argued that the pairing state is independent to the detail LC pattern, so long as the pathway is fixed, i.e., V-V or V-Sb. It would be great if the authors can plot and compare J_q in Eq. (2) or $U_s(k^*, k)$ in Fig. 4 obtained from different LC patterns.

5) There is a typo in the caption for Fig. 1b. The LCs on V-V and V-Sb are shown by purple and blue arrows.

(Remarks on code availability)

Reviewer #2

(Remarks to the Author)

The manuscript by Schultz et al. investigates superconductivity in AV₃Sb₅ “mediated” by soft loop-current (LC) fluctuations. By integrating out the LC fluctuations, the authors derive a current-current interaction, which is then projected into the Cooper channel to solve the linearized BCS gap equation (near T_c) and identify possible pairing symmetries. They report two types of superconducting pairing arising from two distinct LC fluctuations, which they associate with the two superconducting domes observed in CsV₃Sb₅ under pressure. However, I do not recommend publication in Nature Communications for the following reasons:

1. Physically, both LC order/fluctuations and Cooper pairing originate from electronic correlations (if phonons are not considered). That is, both orders are consequences of the bare electron interactions. There is no compelling reason to retain only the current-current interaction in an effective theory, unless interactions in all other channels are negligible. Although the authors cite previous works on cuprates [89–93] to support their approach, this strategy is not widely accepted in the field.
2. Technically, the authors obtain the current-current interaction by integrating out the “LC boson.” However, the resulting interaction should be retarded, especially for a “soft LC boson.” Their interaction, given in Eq. (2), is instantaneous, which implies an infinite LC boson energy—contrary to the assumption of a soft mode. More seriously, a soft LC boson must include damping, analogous to the Millis–Monien–Pines model used for spin fluctuations in cuprates.
3. Another concern is whether the current-current interaction respects gauge invariance. At the bare level, such interactions are generally not gauge-invariant, and it is unclear how gauge symmetry is preserved in their effective theory.

(Remarks on code availability)

Reviewer #3

(Remarks to the Author)

(Remarks on code availability)

Version 1:

Reviewer comments:

Reviewer #1

(Remarks to the Author)

The authors have carefully addressed our comments and provided detailed explanations, along with comprehensive data and additional figures. I have no further concerns and recommend the manuscript for publication.

(Remarks on code availability)

Reviewer #2

(Remarks to the Author)

My main concern still remains: whether LC fluctuation is a good starting point to study the Kagome metals. The authors elaborate this by citing another microscopic calculation based on nearest neighbor Coulomb interaction (PRL 132, 146501, 2024). Then, I don't understand why not starting from that microscopic model directly but the present phenomenological LC fluctuation model? What's the advantage of the present study?

Regarding to the so-called RPA calculation, it actually is not the standard RPA framework [e.g. Scalapino et al. PRB 34, 8190 (1986)], but keeping only the current-current interaction, which only renormalizes itself. In this sense, the comparison with RPA only “makes a show” but not truly justifies the linearized gap equation approach.

Overall, I agree that the calculations and results are solid starting from their model and using their method. But the question is whether this reaches the high standard of Nature Communications and here I have reservations.

(Remarks on code availability)

Reviewer #3

(Remarks to the Author)

(Remarks on code availability)

Response to the comments made by Reviewer #1

In this manuscript, “Superconductivity in kagome metals due to soft loop-current fluctuations”, the authors illustrate that soft fluctuations of translation symmetry-breaking loop currents (LCs) provide a mechanism for unconventional superconductivity (SC) in kagome metals AV₃Sb₅. They argue that the fluctuations of LCs with different pathways lead to two distinct pairing states, with V-V LCs favor chiral $d + id$ state and V-Sb LCs favor s^\pm state. The theory provides a natural explanation for the experimental phase diagram of CsV₃Sb₅ under pressure where a Lifshitz transition takes place at ~ 7.5 GPa, beyond which the Sb-dominant Γ -pocket is removed.

This work presents a comprehensive theoretical framework for the unconventional superconductivity in kagome metals, and emphasizes the crucial role of multi-orbital physics, particularly the role of Sb 5p orbitals. The approach is standard and well-established in the field, and the statements and arguments made in the manuscript are valid and supported by reasonable experimental evidence. It is a promising contribution to the field of kagome superconductivity and would stimulate significant discussion and further research. I would recommend publication in Nature Communications after the authors properly address the following concern.

Our reply: We appreciate the overall positive evaluation of our results and address in what follows the itemized issues brought up by the referee.

1) The LC susceptibility employed in Eq. (2) is phenomenological. While this manuscript emphasizes the crucial role of multi-orbital physics in capturing the correct low-energy physics and pairing instabilities, the multi-orbital physics is not implemented in the phenomenological form of the LC susceptibility. An investigation with LC susceptibility computed from the underlying microscopic model would further strengthen the theory.

Our reply: We thank the reviewer for raising this important point. In the revised manuscript, we now present a microscopic calculation of the loop-current fluctuation propagator starting from the full multi-orbital Hamiltonian. The analysis is performed within the random-phase approximation in the loop-current channel. We demonstrate explicitly that the resulting propagator closely reproduces the phenomenological form used in the original manuscript and leads to the same pairing interaction structure; see Fig. 1 of this reply or the supplementary material. We show plots where we compare the propagators, gap functions, and phase diagrams. In the propagator, we find in the microscopic analysis some weak incommensurability due to imperfect nesting, but the dominant pairing states and their connection to the microscopic loop current pathway are reproduced in detail. In the revised manuscript, we mainly focus on the new results. Additional details are included in the revised Supplementary Information.

Figure 1: In response to question # 1 by referee # 1, we show a table comparing the phenomenological propagator with the RPA propagator. In panels (a),(b), it is the propagator peaking at M_1 . Similarly for panels (c),(d) the peak is at M_2 and for panels (e),(f) the peak is at M_3 . The phenomenological propagator is peaked exactly at the M -points, in accordance with the translation symmetry breaking observed experimentally in the charge ordered state. In contrast, due to imperfect nesting, there is a weak incommensurability in the RPA propagator, because the vanadium states are located slightly away from the M -points. The key assumption of peaking close to the M -points is nonetheless preserved, and hence the pairing interaction is enhanced at this approximate nesting wave vector. In the revised version of the manuscript, we show the phase diagram with this new RPA propagator.

Gap function from phenomenological propagator

Gap function from RPA propagator

Figure 2: In response to question # 1 by referee # 1, we show a table comparing the gap functions obtained using the phenomenological LC propagator with the gap function obtained using the RPA LC propagator. The gap functions in both cases are qualitatively indistinguishable. In panels (a) and (b), we have the E_{2g} gap function, which appears for the fluctuating V-V LC pattern. In panels (c) and (d), we have the s^\pm gap function, which appears for the fluctuating V-Sb LC pattern. The similarity between the gap functions obtained with different approaches justifies our initial assumption of the phenomenological propagator. In the revised manuscript, we compute the phase diagrams with the RPA propagator, and find essentially the same result.

Figure 3: In response to Question #1 raised by Reviewer #1, we present a comparison between the phase diagram from the *original manuscript*, obtained using the phenomenological propagator (left panel) and the *new figure* derived from the propagator calculated within the random phase approximation (right panel). The close quantitative agreement between the two approaches provides *a posteriori* justification for our initial phenomenological model and demonstrates the robustness of our conclusions. In the revised version of the manuscript, we show the results of the new, microscopic analysis requested by Reviewer #1.

2) Are the findings sensitive to the detail electronic structure, Fermi surface topology in particular, of the multi-orbital model? According to DFT calculations [see for example Fig. 2a in PRB 108, L060503 (2023) and Fig. 1b in PRB 111, 235114 (2025)], the two hexagonal Fermi surfaces whose vertices point to the M-point in the BZ are from σ_h -odd orbitals, while the hexagonal Fermi surface whose vertices point to the K-point (the rounded Fermi surface in this work) is due to σ_h -even orbitals. This seems to be little bit different from the Fermi surface topology and orbital content in the multi-orbital model adopted in this work. Since the vertices carry large amount of density of states, could this difference lead to qualitative different results?

Our reply: We appreciate this question and now clarify the extent to which our conclusions depend on the orbital character and geometry of the Fermi surfaces. There does not yet seem to be agreement in the literature about the shape of the Fermi surface due to $\sigma_h = +1$ orbitals. As the referee points out, several DFT studies find the $\sigma_h = +1$ part of the Fermi surface (mostly composed of $d_{3z^2-r^2}$, $d_{x^2-y^2}$ and d_{xy}) with vertices pointing towards K, K' . On the other hand, some other DFT and ARPES studies appear to observe these vertices pointing towards the M -points (a few such studies are listed below). Although the precise shape of the $\sigma_h = +1$ Fermi surface is not settled, it is thankfully not important for our calculation, because the electrons which are believed to be important in superconductivity and charge ordering are primarily $\sigma_h = -1$ (vanadium d_{zx}, d_{yz} , and planar Sb p_z), and hence we focus our calculations on this subset of orbitals.

- CsV₃Sb₅ studies which have the $\sigma_h = +1$ Fermi surface pointing towards the M -points
 - Fig. 3a PRL 125, 247002 (2020)
 - Fig. 1b & 1d Supercond.Sci.Technol. 37 123001 (2024),
 - Fig. 1c Nat Commun 14, 3819 (2023)
- KV₃Sb₅ studies which have the $\sigma_h = +1$ Fermi surface pointing towards the M -points
 - Fig. 1e & 1f Nat Commun 13, 273 (2022)
 - Fig. 1c & 1d Commun Mater 3, 30 (2022)

3) In Fig. 3, the authors sketched the representative LC patterns with ordering wave vector M_3 . It might be helpful for the readers if the authors can show, somewhere in the supplemental, the combined LC patterns from all three ordering wave vectors, and make connections to the V-V LC patterns proposed in previous works, i.e., the pattern proposed in Science Bulletin 66, 1384 (2021) and the four patterns obtained by mean-field calculations in PRB 107, 045127 (2023).

Our reply: We thank the reviewer for the suggestion. In the revised manuscript, we now provide figures illustrating the LC patterns obtained taking a superposition of the patterns with three M -point ordering vectors. We also comment on their relation to previously discussed V-V LC textures. To comment on the relationship between our patterns and the works referenced by the reviewer:

- The pattern in Fig. 2c of Science Bulletin 66, 1384 (2021) is in the irrep mM_2^+ , and is a weighted superposition of our patterns in our (referee response letter figs) Figs. 4d and 5d below.
- The four patterns in Fig. 4b-4e of PRB 107, 045127 (2023) are all in the irrep mM_2^+ . In particular, they are all weighted superpositions of our patterns in our (referee response letter figs) Figs. 4d and 5d below, and their Fig. 4d even appears to be the specific linear combination comprising only our pattern in our (referee response letter figs) Fig. 5d below.
- We note that our patterns in (referee response letter figs) Figs. 6 (inversion even V-Sb), 7 (inversion odd V-V), and 8 (inversion odd V-Sb and V-V) are not represented in the two works cited by the referee.

Figure 4: In response to referee # 1 question # 3, we include the rotated counterparts of the LC pattern from main text Fig. 3a. This set of patterns is inversion even, and only flows between V-V. Panel (a) has ordering wave vector \mathbf{M}_1 , panel (b) has ordering wave vector \mathbf{M}_2 , and panel (c) has ordering wave vector \mathbf{M}_3 . Panel (d) is the equal weight superposition of panels (a),(b),(c), which maintains C_{6z} symmetry. The panels (a),(b),(c) are related to each other by $\pi/3$ rotation.

Figure 5: In response to referee # 1 question 3, we include the rotated counterparts of the LC pattern from main text Fig. 3b. This set of patterns is inversion even, and only flows between V-V. Panel (a) has ordering wave vector \mathbf{M}_1 , panel (b) has ordering wave vector \mathbf{M}_2 , and panel (c) has ordering wave vector \mathbf{M}_3 . Panel (d) is the equal weight superposition of panels (a),(b),(c), which maintains C_{6z} symmetry. The panels (a),(b),(c) are related to each other by $\pi/3$ rotation. The pattern in panel (d) appears to be the one in Fig. 4d of the work PRB 107, 045127 (2023).

Figure 6: In response to referee # 1 question 3, we include the rotated counterparts of the LC pattern from main text Fig. 3c. This set of patterns is inversion even, and flows between V-Sb. Panel (a) has ordering wave vector \mathbf{M}_1 , panel (b) has ordering wave vector \mathbf{M}_2 , and panel (c) has ordering wave vector \mathbf{M}_3 . Panel (d) is the equal weight superposition of panels (a),(b),(c), which maintains C_{6z} symmetry. The panels (a),(b),(c) are related to each other by $\pi/3$ rotation.

Figure 7: In response to referee # 1 question 3, we include the rotated counterparts of the LC pattern from main text Fig. 3d. This pattern is inversion odd, and only flows between V-V. Panel (a) has ordering wave vector \mathbf{M}_1 , panel (b) has ordering wave vector \mathbf{M}_2 , and panel (c) has ordering wave vector \mathbf{M}_3 . Panel (d) is the equal weight superposition of panels (a),(b),(c), which only has C_{3z} symmetry due to the inversion breaking. The panels (a),(b),(c) are related to each other by $\pi/3$ rotation.

Figure 8: In response to referee # 1 question 3, we include the rotated counterparts of the LC pattern from main text Fig. 3e. This pattern is inversion odd, and flows both between V-V and V-Sb. Panel (a) has ordering wave vector \mathbf{M}_1 , panel (b) has ordering wave vector \mathbf{M}_2 , and panel (c) has ordering wave vector \mathbf{M}_3 . Panel (d) is the equal weight superposition of panels (a),(b),(c), which only has C_{3z} symmetry due to the inversion breaking. The panels (a),(b),(c) are related to each other by $\pi/3$ rotation.

4) It is argued that the pairing state is independent of the detail LC pattern, so long as the pathway is fixed, i.e., V-V or V-Sb. It would be great if the authors can plot and compare J_q in Eq. (2) or $U_s(k^*, k)$ in Fig. 4 obtained from different LC patterns.

Our reply: We thank the reviewer for the suggestion, and based on the suggestion we have updated the supplementary with figures of the projected pairing kernel for all LC patterns, as well as added a further discussion of the intuition behind the pairing symmetries. We think that readers will obtain a better intuition for the results due to the inclusion of these figures. We show here, in the referee report, plots which support the intuition that, for all LC patterns, either $d + id$ (for V-V LC) or s_{\pm} (for V-Sb LC) are obtained. Combined with our linearized gap equation calculation, *we are able to conclude that the pairing symmetry depends fundamentally on the microscopic loop current pathway, as this strongly determines the Fermi surface regions that feel the largest interaction.*

It is worth mentioning some specific details. Looking at this (referee response figure) Fig. 9 for the 5 types of LC patterns, we make the following observations:

- If we compare the even vs. odd parity V-V loop current patterns, e.g. Fig. 9(a) vs. (j), we first see that there is still nearly no interaction to the Γ -pocket, and hence no s^{\pm} pairing.
- Again comparing even vs. odd parity V-V loop current patterns Fig. 9(a) vs. (j), we see in (a) there is a strong interaction between M -points. However, this specific interaction is suppressed in (j) because of the odd parity of the current pattern. In spite of this, there is strong interaction to wave vectors close to the M -point, although not right at the M -point. This makes the leading pairing symmetry, $d + id$, less obvious than in the even parity case. Nonetheless, our calculations show that $d + id$ prevails in both the even and odd parity V-V LC patterns.
- We can also compare the two different patterns which are V-Sb, one of which is parity even, and the other parity odd. In particular, if we examine Fig. 9(g) vs. (m), we see that there is a large interaction between the outer Fermi surface and the inner Fermi surface. This fact is true regardless of the parity, and tells us that the favoured pairing symmetry is likely to be s^{\pm} , which is indeed what is found through the linearized gap equation calculation.

We further emphasize this point in the supplementary, wherein these additional projected pairing kernels are shown.

Figure 9: Singlet interactions. (a)-(c) are for main text Fig. 3a. (d)-(f) are for main text Fig. 3b. (g)-(i) are for main text Fig. 3c. (j)-(l) are for main text Fig. 3d. (m)-(o) are for main text Fig. 3e. The key feature is that the first, second, and fourth rows (which are from V-V LC patterns) have very small interaction with the Γ pocket, whereas lines 3 and 5 (which have a V-Sb component) have strong interaction with the Γ pocket. This is the reason why the pathway is the determining factor for the resulting pairing symmetry.

5) There is a typo in the caption for Fig. 1b. The LCs on V-V and V-Sb are shown by purple and blue arrows.

Our reply: We thank the reviewer for catching this error; the caption of main text Fig. 1b has been corrected.

Response to the comments made by Reviewer #2

The manuscript by Schultz et al. investigates superconductivity in AV_3Sb_5 “mediated” by soft loop-current (LC) fluctuations. By integrating out the LC fluctuations, the authors derive a current-current interaction, which is then projected into the Cooper channel to solve the linearized BCS gap equation (near T_c) and identify possible pairing symmetries. They report two types of superconducting pairing arising from two distinct LC fluctuations, which they associate with the two superconducting domes observed in CsV_3Sb_5 under pressure. However, I do not recommend publication in Nature Communications for the following reasons:

Our reply: We thank the reviewer for the report, and provide discussion below and in the manuscript justifying the reasoning behind our approach, in light of the complicated multi-orbital Fermi surface and conceptual clarity afforded by the group theoretic analysis.

1. Physically, both LC order/fluctuations and Cooper pairing originate from electronic correlations (if phonons are not considered). That is, both orders are consequences of the bare electron interactions. There is no compelling reason to retain only the current-current interaction in an effective theory, unless interactions in all other channels are negligible. Although the authors cite previous works on cuprates [89–93] to support their approach, this strategy is not widely accepted in the field.

Our reply: It is indeed true that the microscopic investigation of loop-current order and fluctuations is less developed than more conventional states of order. However, we feel that isolating the effects of loop current fluctuations, especially group theoretically, provides a remarkably simple intuition in the solution of the problem, and that this clarity is a significant asset of our theory. We also emphasize that the analysis itself is based on technical tools that are firmly established in the literature. The steps parallel to a significant extent what was done to analyze spin- or bond-charge-density wave induced superconductivity. Hence, we agree with the referee that superconductivity due to loop-current fluctuations is a comparatively unexploited pairing mechanism, but the method to establish it is in our view widely accepted. Furthermore, in PRL 132, 146501 (2024) it was demonstrated that for a significant range of microscopic parameters, the leading instability of a model for the kagome metals is indeed a loop-current ordered state. Our theory directly builds on these insights. In addition, we feel that the approach of isolating loop current fluctuations provides conceptual clarity, especially because the group theoretic analysis yields transparent interpretation of the results. In the revised version of the manuscript, we have explicitly performed an analysis of the microscopic loop-current propagator and demonstrated that it is a sharply peaked function in momentum space.

2. Technically, the authors obtain the current-current interaction by integrating out the “LC boson.” However, the resulting interaction should be retarded, especially for a “soft LC boson.” Their interaction, given in Eq. (2), is instantaneous, which implies an infinite LC boson energy—contrary to the assumption of a soft mode. More seriously, a soft LC boson must include damping, analogous to the Millis–Monien–Pines model used for spin fluctuations in cuprates.

Our reply: The inclusion of retardation effects in strongly correlated electron systems is an interesting and important issue. It has in the past been investigated in great detail, including by some of us. Such effects are most essential when one is near a quantum critical point (QCP), where the dynamics of order-parameter fluctuations can under no circumstances be ignored. The situation becomes easier as one moves away from a QCP or, what seems to be appropriate for the kagome systems, if the transition is in fact weakly of first order. Then these retardation effects can be ignored in the weak coupling limit, in just the same way as was done in the BCS theory of the retarded electron-phonon interaction. This perspective was used in PRL 114, 097001 (2015) for nematic systems and in Sci. Adv. 10, eadn3662 (2024) for translation-invariant loop current fluctuations. We emphasize that the complicated nature of the Fermi surface of the kagome superconductors, with a total of 13 orbitals per unit cell, the chosen path of starting our investigation in the weak coupling limit seems to be the only viable one that enables the exploration of a wide parameter range and allows for a transparent physical interpretation. Maybe most importantly, in all cases known to us, the additional inclusion of retardation effects did not change the pairing symmetry that was first obtained in a weak coupling approach. In the revised version, we have clarified the reasons for our approach.

3. Another concern is whether the current-current interaction respects gauge invariance. At the bare level, such interactions are generally not gauge-invariant, and it is unclear how gauge symmetry is preserved in their effective theory.

Our reply: We thank the reviewer for this question, and we agree that the gauge invariance of the theory is an important point. In the absence of an external gauge field, gauge invariance is essentially a restatement of charge

conservation. The Hamiltonian is clearly invariant under global U(1) transformations. However, the case of local charge conservation is indeed more subtle. To check local charge conservation, one needs to verify that

$$\frac{d}{dt}\langle\hat{n}_i\rangle=0 \tag{1}$$

$$\langle[\hat{H},\hat{n}_i]\rangle=0 \tag{2}$$

$$\sum_j\langle\hat{j}_{ij}\rangle=0 \tag{3}$$

If we enter the ordered phase by allowing the loop current order parameter to condense, thereby breaking time-reversal symmetry, then $\sum_j\langle\hat{j}_{ij}\rangle$ needs to be analyzed carefully. In our analysis we have explicitly checked and ensured that all loop current patterns respect Kirchhoff's law, i.e. the local continuity equation. This can be seen qualitatively in the LC patterns depicted in main text Fig. 3, and the quantitative statement is in Table 1 of the Methods section. We add that the importance of carefully analyzing the issue of the local continuity equation was discussed earlier in Sci. Adv. 10, eadn3662 (2024) by some of us. Alternatively, in the fluctuating phase (where the loop current boson has been integrated out), there is no possibility of the violation of local charge conservation, because the interaction was generated through a Kirchhoff law-respecting current pattern. This clarifies local charge conservation in the fluctuating phase. We have updated the main text to clarify this point.

Response to the comments made by Reviewer #3

Our reply: We thank Reviewer #3 and their co-reviewer for their time and for participating in the Nature Communications co-review initiative.

Response to the comments made by Reviewer #1

The authors have carefully addressed our comments and provided detailed explanations, along with comprehensive data and additional figures. I have no further concerns and recommend the manuscript for publication.

Our reply: We thank referee # 1 for recommending our work for publication.

Response to the comments made by Reviewer #2

My main concern still remains: whether LC fluctuation is a good starting point to study the Kagome metals. The authors elaborate this by citing another microscopic calculation based on nearest neighbor Coulomb interaction (PRL 132, 146501, 2024). Then, I don't understand why not starting from that microscopic model directly but the present phenomenological LC fluctuation model? What's the advantage of the present study?

Regarding to the so-called RPA calculation, it actually is not the standard RPA framework [e.g. Scalapino et al. PRB 34, 8190 (1986)], but keeping only the current-current interaction, which only renormalizes itself. In this sense, the comparison with RPA only "makes a show" but not truly justifies the linearized gap equation approach.

Overall, I agree that the calculations and results are solid starting from their model and using their method. But the question is whether this reaches the high standard of Nature Communications and here I have reservations.

Our reply: We thank the reviewer for the report, and provide discussion below and in the manuscript justifying the reasoning behind our approach, in light of the complicated multi-orbital Fermi surface and conceptual clarity afforded by the group theoretic analysis.

We thank the referee for raising important conceptual questions. We are encouraged that the referee finds the calculations and results to be solid within our framework. Below we address the main concerns in detail.

1. **On the relation to RPA and the linearized gap equation:** The referee comments that our treatment differs from the "standard" RPA formulation (e.g., Scalapino et al., PRB 34, 8190 (1986)) and suggests that the comparison may not fully justify the use of the linearized gap equation. We appreciate the opportunity to clarify this point. The standard RPA framework developed in the context of single-band Hubbard models focuses on local interactions (on-site U) and their decomposition into spin and charge channels. In contrast, Kagome metals are inherently multi-orbital systems with significant inter-site and inter-orbital structure. As a result, the relevant interaction channels are qualitatively different and include loop-current (LC) fluctuations. In our treatment we first identify the relevant interaction channel associated with LC fluctuations and then perform a channel decomposition appropriate for this multi-orbital, non-local interaction structure. Within this framework, the effective interaction in the LC channel, and any other channel for that matter, renormalizes itself, which is the standard feature of RPA-type resummations. Thus, while our formulation differs in implementation from the original single-band Hubbard RPA of Scalapino et al., it is conceptually fully consistent with the RPA philosophy: namely, the resummation of fluctuations in a given channel and their feedback on pairing.
2. **On the choice of loop-current (LC) fluctuation model vs. starting from a microscopic interaction:** The referee raises the question why we adopt a phenomenological LC fluctuation framework rather than starting directly from a microscopic model such as that studied in Ref. [PRL 132, 146501 (2024)]. First, we wish to stress that with the revised version of our manuscript we do not consider our approach phenomenological. Loop-current fluctuations have, instead, been determined from a microscopic theory. Second, we would like to clarify the distinction in scope and objective between our work and the above reference. The cited work demonstrates that nearest-neighbor Coulomb interactions on the Kagome lattice can stabilize loop-current (LC) order at the mean-field level. However, addressing superconductivity requires going beyond the establishment of ordered states and, crucially, analyzing fluctuations of the corresponding order parameter. Our work is precisely aimed at this next step. We analyze LC fluctuations as the relevant low-energy degrees of freedom and investigate their role as a pairing glue. This is conceptually analogous to spin-fluctuation theories of superconductivity, where establishing magnetic order does not suffice to identify the pairing state. Instead one has to develop a theory of magnetic correlations and fluctuations to solve the pairing problem.

We acknowledge the referee's concern regarding the scope of the work. We would like to emphasize that our study addresses a timely and actively debated question: the nature of pairing mechanisms in Kagome metals, where unconventional orders and multi-orbital effects play a central role. Our contribution is i) to establish a concrete and physically motivated fluctuation mechanism (LC fluctuations) for superconductivity in these systems. ii) to provide a

systematic and transparent theoretical framework to analyze its consequences, and iii) to bridge microscopic proposals of LC order with experimentally relevant pairing phenomena.

Response to the comments made by Reviewer #3

Our reply: We thank Reviewer #3 and their co-reviewer for their time and for participating in the Nature Communications co-review initiative.